Object-oriented building extraction based on visual attention mechanism

Shen Xiaole
Yu Chen
Lin Lin
http://orcid.org/0000-0001-6201-3251 Cao Jinzhou caojinzhou@sztu.edu.cn
College of Big Data and Internet, Shenzhen Technology University , Shenzhen, Guangdong , China
Wang Jingzhe
Electronic publication date: 2023 Aug 30
Publication date: 2023
Volume: 9
Electronic Location ID: e1566
Received 2023 Jul 6; Accepted 2023 Aug 13
Copyright: © 2023 Shen et al.
Copyright year: 2023
Copyright holder: Shen et al.
License: This is an open access article distributed under the terms of the Creative Commons Attribution License, which permits unrestricted use, distribution, reproduction and adaptation in any medium and for any purpose provided that it is properly attributed. For attribution, the original author(s), title, publication source (PeerJ Computer Science) and either DOI or URL of the article must be cited.
License URL: https://creativecommons.org/licenses/by/4.0/

Keywords: Building extraction, Visual attention, Shadow extraction, Object-oriented

Funding: National Natural Science Foundation of China 42001393, 41501370 and 62176165 5th College-enterprise Cooperation Project of Shenzhen Technology University 2021010802014 Shenzhen Science and Technology Program JCYJ20220530152817039 Guangdong Science and Technology Strategic Innovation Fund Guangdong–Hong Kong-Macau Joint Laboratory Program 2020B1212030009 This research was supported by the National Natural Science Foundation of China (Grant Nos. 42001393, 41501370 and 62176165), the 5th College-enterprise Cooperation Project of Shenzhen Technology University (Grant No. 2021010802014), Shenzhen Science and Technology Program (Grant No. JCYJ20220530152817039), and Guangdong Science and Technology Strategic Innovation Fund (the Guangdong–Hong Kong-Macau Joint Laboratory Program, Grant No. 2020B1212030009). The funders had no role in study design, data collection and analysis, decision to publish, or preparation of the manuscript.

==============================
Buildings, which play an important role in the daily lives of humans, are a significant indicator of urban development. Currently, automatic building extraction from high-resolution remote sensing images (RSI) has become an important means in urban studies, such as urban sprawl, urban planning, urban heat island effect, population estimation and damage evaluation. In this article, we propose a building extraction method that combines bottom-up RSI low-level feature extraction with top-down guidance from prior knowledge. In high-resolution RSI, buildings usually have high intensity, strong edges and clear textures. To generate primary features, we propose a feature space transform method that consider building. We propose an object oriented method for high-resolution RSI shadow extraction. Our method achieves user accuracy and producer accuracy above 95% for the extraction results of the experimental images. The overall accuracy is above 97%, and the quantity error is below 1%. Compared with the traditional method, our method has better performance on all the indicators, and the experiments prove the effectiveness of the method.

Introduction

In recent years, with the global launch of high-resolution RSI satellites and the popularization of aerial remote sensing, especially UAV remote sensing technology, the acquisition means of high-resolution RSI and very high-resolution RSI are richer, and the acquisition difficulty is lower, and cost is lower. With the development of image processing and analysis technology, how to extract various buildings in RSI has gradually become a research hotspot.

The method for extracting buildings is based on their inherent features, which include both geometric and spectral properties. Buildings typically have regular shapes, often consisting of one or more rectangles. Therefore, the extraction process involves identifying straight lines and their combinations within the image to accurately extract the buildings (Ahmadi et al., 2010; Mayunga, Coleman & Zhang, 2010; Turker & Koc-San, 2015). On the other hand, due to the diversity of buildings and their roof materials, the spectral features of buildings are not uniform, and there may be significant spectral differences between different buildings. However, buildings generally exhibit higher brightness compared to other ground objects. In high-resolution RSI, there is a pronounced contrast in brightness between buildings and their surrounding environment, which gives rise to the distinctive spectral and structural features of buildings (Huang & Zhang, 2011; He & Wang, 2018).

Ahmadi et al. (2010) proposed a new active contour model for extracting the edges of buildings. They utilized this model to extract the contours of buildings. Mayunga, Coleman & Zhang (2010) used the radial casting method to initialize snakes contours and accurately extract the contours of buildings using this snakes model. Turker & Koc-San (2015) employed the SVM method to generate candidate building blocks. They then performed Canny edge detection and Hough transform on each image block to extract line segments. Finally, they utilized perceptual grouping to extract the contours of buildings. Aytekin et al. (2009) used high spatial resolution panchigh-resolution omatic imagery and lower spatial resolution multispectral data for building extraction. The extraction process initially classified the fused imagery into natural and man-made areas using the normalized difference vegetation index (NDVI). Cheng & Han (2016) achieved good results in building extraction by combining additional information such as shadows and lighting with higher-level features such as HOG and texture. Dikmen & Halici (2014) first identified shadow objects in over segmented imagery and then merged adjacent suspected building shadow objects.

In addition to the aforementioned traditional methods, there have been recent advancements in utilizing deep learning techniques for extracting buildings. Wang, Sun & Zhao (2021) proposed a multi-scale feature fusion and alignment network (MFANet) that utilizes the structure of an encoder, decoder, and multi-level feature fusion blocks to effectively extract and fuse multi-scale and global features, thereby improving segmentation results. Additionally, Zuo et al. (2021) introduced a network called hybrid network with multi-attention mechanism (HMANet), which captures spatial, channel, and category information adaptively to model and calibrate global correlations. In another work, Li et al. (2022) presented an improved semantic segmentation framework that incorporates graph reasoning and disentanglement learning to address segmentation challenges and further enhance segmentation accuracy. Pastorino et al. (2022) proposed a method that combines deep learning with probabilistic graphical models to process and classify multi-scale remote sensing data by integrating hierarchical Markov models. Zhao et al. (2021) introduced a semantic segmentation network called SSAtNet, which is based on an end-to-end attention mechanism. By incorporating pyramid attention pooling, this network introduces adaptive feature refinement to effectively correct detail information.

Different from medium and low resolution RSI, in high-resolution RSI and very high-resolution RSI, the details of ground objects are clearer and the information is richer, and the phenomena of same object different spectrum and foreign object same spectrum are more common. Therefore, the traditional ground feature extraction method using only the low-level features of RSI has been unable to meet the application requirements of building extraction.

We make full use of spectral features, shape features, texture features, edge features, context and topological relations of buildings in high-resolution RSI. We propose an object-oriented method for high-resolution RSI shadow extraction. Based on visual attention mechanisms based on information theory and the object-oriented image analysis method, we propose an object-oriented high-resolution RSI building extraction method under visual attention mechanism. Our method combines bottom-up RSI low-level feature extraction with top-down guidance from prior knowledge.

Methods

Construction of a knowledge base for the shadow object feature

Shadow is ubiquitous in high-resolution RSI and is one of the basic features of RSI. Shadow is caused by many factors such as illumination conditions, terrain undulation, ground cover, etc. Shadow affects the interpretation of ground objects and brings difficulties to ground object extraction, target recognition, and change detection. Shadows can be generated by specific features such as buildings, trees, mountains, etc. (Zhou & Fu, 2021; Mostafa, 2017).

In various types of RSI, shadows exhibit different features due to the presence of different types of ground objects. Moreover, certain ground objects possess spectral features that resemble those of shadows. Consequently, when relying solely on a single feature to determine a criterion for distinguishing shadows, these ground objects may be erroneously identified as shadows. Hence, it is imperative to consider multiple features when discerning between shadows and other ground objects. This necessitates the extraction of various features to construct an object feature knowledge base and establish a discriminant criterion that accounts for the influence of different ground objects. The features used for constructing this knowledge base are as follows. Brightness

The brightness indicated the light intensity reflected by the ground objects received by the sensor. The shadow area does not receive direct sunlight, and its brightness mainly comes from the scattering light. Because the light intensity of the shadow area is much lower than that of the non-shadow area, the brightness of the shadow area is usually low.

Hue

Due to the influence of Rayleigh scattering, the blue light with shorter wavelength accounts for a large proportion of sky light, which leads to the shift of the spectrum in the shadow area to the blue band, thus the hue becomes higher.

Ratio/difference between green band and blue band

In the visible light band, water has strong absorption of sunlight, and its reflection is mainly in the blue band. However, due to the influence of algae chlorophyll in water, reflection will be formed in the green band. If the suspended sediment content in water is high, the spectral value will shift to the red and green bands. In RSI, water has high hue and low brightness. If only the hue and brightness features are judged, it may lead to false extraction. In addition, some dark green trees also show similar features in RSI.

According to the features of these ground objects and shadows, the ratio or difference between green band and blue band can be used to distinguish them. The ratio or difference of these ground objects is generally greater than that of the shadow area.

Convexity model

From the perspective of brightness, the brightness of shadow areas in remote sensing image is low, and the brightness of surrounding objects is generally higher than that of shadow area, which constitutes a concave convex surface. From the perspective of hue, the hue of the shadow area is high, and the hue of the surrounding objects is generally lower than that of the shadow area, forming a convex surface.

Area and shape

Compared with lakes and other ground objects, the area of shadow area is relatively small. Compared with rivers, ditches and other ground objects, the shadow area is more compact in shape.

After establishing a knowledge base for shadow features, it is still necessary to determine the criteria for each feature. This article utilizes the maximum between-class variance method to adaptively select the threshold and establish the criterion for shadow objects across different features.

Shadow extraction method process

In RGB color space, the shadow region in remote sensing image has no obvious features in red, green, and blue bands. Overall, hue, saturation, and brightness models have more advantages in remote sensing image shadow extraction. To make full use of the brightness, hue, and other features of shadows in RSI, and to use the convex model criterion as an auxiliary criterion to constrain the segmentation object in the image segmentation process, this article uses HSV color space for object-oriented segmentation of images. This article utilizes the object-oriented image analysis method to analyze the features of shadow regions in high-resolution RSI. It also constructs a feature knowledge base of shadow objects and proposes an object-oriented shadow extraction method for RSI. This method is based on multi-scale segmentation in the HSV color space. According to the flowchart provided in Fig. 1, the specific implementation process of the method is as follows:

Figure 1 Flow of shadow extraction method.

The input image undergoes a color space transformation, converting it from RGB color space to HSV color space.

Various features of input images are extracted from both the RGB color space and the HSV color space. These features are then combined, and the adaptive threshold is calculated using the maximum between-class variance method.

Using object-oriented multi-scale segmentation method, combined with convex model criteria to segment image.

Extract the area, shape and convex model of the image object in the segmentation results.

Combine the features obtained from step (2) and step (4) to construct the shadow object feature knowledge base.

By utilizing a knowledge base of shadow features, each image object in the segmentation results is individually assessed based on discriminant criteria to determine if it is a shadow. This process ultimately yields the extraction results.

The building salient index under the visual attention mechanism

To make full use of the color features of buildings in high-resolution RSI and form a greater degree of differentiation with other types of features, this article proposes a remote sensing image feature space transformation method that considers the features of buildings, which is used to describe the spectral features of buildings in high-resolution RSI. Combined with texture features, a building saliency index (BSI) is proposed. Architectural feature dimension

Buildings usually have high brightness in high-resolution RSI. Through the observation of a large number of buildings in high-resolution RSI, this article believes that the color of the building roofs in the image are mainly gray, blue and red, and green roofs are less common. Based on this feature, this article proposes the feature transformation formula of architectural feature dimension:

(1) f1=2×λ1×R+2×(1−λ1)×B−G

(2) F1={f1    if f1>00     if f1<=0

In the formula, R, G, and B are the DN values of the image in the red, green, and blue bands. λ1 is feature weights, which need to satisfy λ1∈ between 0 and 1. When the image is dominated by a reddish roof, λ1 is greater than 0.5; when the image is dominated by blue roofs, λ1 is less than 0.5; when the roof color in the image is gray, λ1 equals 0.5. When the roof color in the image is unknown or there are multiple colors of the roof, Eq. (1) can be replaced by the following equation:

(3) f1=2×max(R,B)−G

Vegetation feature dimension

Vegetation usually has green hue and high saturation in high-resolution RSI. Based on this feature, this article proposes the feature transformation formula of the vegetation feature dimension:

(4) f2=G−λ2×R−(1−λ2)×BG+λ2×R+(1−λ2)×B

(5) F2={f2iff2>00iff2<=0

The vegetation feature dimension highlights the vegetation features in the image, such as grassland, and cultivated land woodland, which can better distinguish high-brightness vegetation and buildings, as well as low-brightness vegetation and shadow. If the obtained RSI contain near infrared bands, the normalized difference vegetation index (NDVI) can be used to replace the vegetation feature dimension. Shadow feature dimension

Shadow has the features of low brightness and high hue in high-resolution RSI. Based on this feature, this article proposes the feature transformation formula of shadow feature dimension:

(6) F3=HV

H and V are hue and brightness components of the image in HSV color space respectively.

The transformed image feature space is comprised of three dimensions: normalized building feature dimension F1, vegetation feature dimension F2, and shadow feature dimension F3. In this feature space, the article proposes a building spectral feature index to better highlight the features of buildings, while suppressing vegetation and shadow features:

(7) IF=2×F1−F2−F32×F1+F2+F3

Considering the spectral and texture features of buildings in high-resolution RSI, this article proposes the building visibility index in high-resolution RSI:

(8) BSI=IF×BASI

The BASI is built-up areas saliency index that indicates the probability of a given location being within a building area (Shao, Tian & Shen, 2014).

The adjacency criterion of buildings and shadows

In RSI, buildings are often accompanied by shadows. High-resolution RSI provides clearer visibility of both buildings and their corresponding shadows compared to low-resolution RSI. Additionally, there is a topological relationship between buildings and their shadows in RSI. In the Northern Hemisphere, when studying a region, the sun’s orientation is typically from south to east or south to west. Consequently, the shadow region in the building area is adjacent to it in the direction of north to west or north to east. The utilization of topological relations between buildings and building shadows aids in differentiating buildings from objects that share similar spectral features, such as squares and roads. This article outlines the adjacent criterion for buildings and shadows as follows. For a building candidate object Objb, there is an image object Objs. If the following conditions are satisfied, Objb and Objs conform to the adjacency criterion, and object Objb can be judged as a building object.

(9) SHADOW(Objs)=TRUE

SHADOW is a logical property defined on an image object. The above equation represents an object Objs as a shadow object.

(10) ADJ(Objb,Objs)=TRUE

ADJ is a logical attribute defined between any two image objects. The above equation indicates that there is an adjacent topological relationship between object Objb and object Objs.

(11) 1nb∑(x,y)∈Objby−1ns∑(x,y)∈Objsy<0

nb and ns are the number of pixels contained in Objb and Objs, respectively. The above equation indicates that the center of gravity of object Objb is lower than that of object Objs in the longitudinal direction. For the image products with coarse correction, the longitudinal axis in the image is usually in the north-south direction. The above equation considers that the adjacent shadow objects should be above the candidate building objects.

(12) ∑(x,y)∈Objb(x,y+1)∈Objs1>r×(∑(x,y)∈Objb(x−1,y)∈Objs1+∑(x,y)∈Objb(x+1,y)∈Objs1)

The ratio parameter r is greater than 0, and usually r is equal to 1. The above equation indicates that in the adjacent edge points of object Objb and Objs, the case that the edge point of Objb is close to the edge point of Objs is more than the case that the edge point of Objb is close to the edge point of Objs. This article believes that under normal circumstances, the main direction of the adjacent edge of the building shadow and the building should be horizontal. The above equation can eliminate some pseudo-building objects generated by other ground object shadows. For most multi-story and middle-high buildings in the north-south direction, The above equation is applicable. For some buildings in the east-west direction, it can be extracted by setting the ratio parameter r less than 1.

Flow of building extraction method

According to the flowchart provided in Fig. 2, the specific implementation process of the method is as follows:

Figure 2 Flow of building extraction method.

1. Remote sensing image preprocessing

For RSI with uneven illumination, a color consistency processing method is used for image preprocessing. For RSI with thin cloud effect, the uniform light method can be used for preprocessing. The remote sensing image obtained after preprocessing is clearer and the details are richer, which is conducive to the subsequent processing of images such as segmentation and feature extraction. 2. Main feature extraction

The RGB color space of the pre-processed RSI is converted to a new feature space using a method specifically designed for RSI, taking into account the unique features of buildings. This transformation allows for distinct features of buildings, shadows, and vegetation to be observed in different dimensions within the new feature space, which facilitates the subsequent extraction of shadows and buildings. Additionally, texture features and edge features of the image are extracted, resulting in a composite of the main features of the image, including building features, vegetation features, and shadow features. 3. Visual saliency computation

According to the features of building areas and buildings in high-resolution RSI, the top-down prior knowledge is introduced to construct the texture features of building areas, and the visual attention model based on information theory is used to calculate the original index BASI of building areas. Then, according to the main features extracted in Step (2), the spectral feature index of the building is calculated, and combined with BASI, the building visible index BSI is calculated. 4. Object-oriented multi-scale segmentation

Based on the object-oriented image analysis method, the object-oriented remote sensing image multi-scale segmentation method is used to segment the remote sensing image into the transformed feature space, and the multi-scale image object set is obtained. 5. Rough extraction of buildings

According to the segmentation results of step (4), select the appropriate scale, and use the following formula to count the building features one by one for the image objects of this scale.

(13) Qbuilding(Ok)=1nk∑(i,j)∈OkBSI(i,j)

Ok is the kth image object obtained by segmentation, Qbuilding(Ok) represents the building features of object Ok, nk represents the total number of pixels contained in object Ok, and BSI(i,j) represents the building salient index at the midpoint (i,j). According to the statistical results of building features of all objects, whether the object is a candidate building object is judged by a given threshold. The maximum between-class variance method is used to obtain the adaptive threshold. 6. Building area extraction

The object-oriented building area extraction method under the visual attention mechanism was used to extract the building area of RSI in the transformed feature space according to the building area dominant index BASI calculated in Step (3) (Shao, Tian & Shen, 2014). 7. Shadow extraction

Using the object-oriented remote sensing image shadow extraction method, the shadow object feature knowledge base is established. According to the segmentation results of Step (4), the appropriate scale is selected to extract the shadow of the image and obtain the set of shadow objects.

When the solar elevation angle at the time of image acquisition is large, the shadow area is usually smaller than the building area, and then the shadow object can be extracted at a smaller scale. When the solar elevation angle at the time of image acquisition is small, the shadow area is similar is size to the building area, and the shadow object can be extracted at the same scale. 8. Fine extraction of buildings

The building object set Rbuilding should have the following relationship with the building area object set Rbuild−up:

(14) Rbuilding⊂Rbuild−up

This article argues that buildings should be inside the built-up area. Using the adjacency criterion of buildings and shadows proposed in the thesis, and combining the inclusion criterion of buildings and building areas, the pseudo-building objects are eliminated and the building objects are obtained.

Finally, a morphological closed operation and an open operation are performed on the binary image of the extraction results to eliminate the “burr” in the extraction results and obtain the final building extraction results.

Experiment

Shadow extraction experiments

To validate the effectiveness of the method, this article selected two high-resolution remote sensing images for shadow extraction experiments. Additionally, manual extraction of shadow regions from the remote sensing images was performed through visual interpretation and used as reference extraction results. Two contrasting experiments were conducted using the shadow extraction methods based on spectral ratio (Tsai, 2006) and continuous thresholding (Chung, Lin & Huang, 2009). In order to compare the differences in extraction results between pixel-level and object-level methods, the spectral ratio method was also applied in the HSV color space.

Experiment 1 selected an aerial remote sensing image of the Beijing region, as shown in Fig. 3A. The manually extracted reference shadow region is shown in Fig. 3B, with the black portion representing the shadow area. Figures 3C and 3D display the extraction results of the spectral ratio-based method and the continuous threshold-based method, respectively. Figure 3E illustrates the shadow extraction results of the proposed method in this article.

Figure 3 Experiment 1 comparison of shadow extraction results.

In the following experimental results, red represents true positives (TP), blue represents false positives (FP), and green represents false negatives (FN). TP, FP and FN are explained in the Evaluating Indicator section.

Comparing the extraction results of three methods, it can be observed that the methods based on spectral ratio and continuous thresholding fail to fully extract some building shadows, resulting in missed extractions. Additionally, these methods do not employ object-oriented image analysis methods, leading to the presence of salt-and-pepper noise in the extraction results. On the other hand, the method proposed in this article provides more complete extraction of shadow areas and successfully captures all shadows of multi-story buildings. However, the article’s method fails to accurately extract the shadows of a row of trees projecting onto the grass in the lower left area of the image. This is because the area of these tree shadows in the image is too small, and the article’s method tends to prioritize the accurate extraction of larger building shadows. The segmentation scale used in the method is larger than the size of the tree shadows, resulting in the tree shadows not being segmented as independent objects.

Experiment 2 selected a part of WorldView-2 satellite remote sensing image near Wuhan Wuhan University, as shown in Fig. 4A. The manually extracted reference shadow area is shown in Fig. 4B, where the black part represents the shadow area. Figures 4C and 4D show the extraction results of the method based on spectral ratio and the method based on continuous threshold, respectively. Figure 4E shows the shadow extraction results of the proposed method.

Figure 4 Experiment 2 comparison of shadow extraction results.

Comparing the extraction results of three methods, it can be observed that the method based on spectral ratio and the method based on continuous threshold mistakenly extract a portion of deep green trees as shadows. Furthermore, due to the lack of an object-oriented image analysis approach, these methods result in a salt-and-pepper effect. The method proposed in this article successfully extracts complete building shadows, but it fails to accurately extract some tree shadows. This is because sunlight penetrates through the trees to a certain extent, causing the shadow areas to shift towards the green band. Additionally, tree shadow areas are usually scattered and have smaller individual areas. The method in this article tends to accurately extract larger building shadows, and the chosen segmentation scale fails to segment some tree shadows as independent objects.

Building extraction experiments

Experiment 3 is an aerial remote sensing image of Beijing, as shown in Fig. 5A. The reference buildings artificially extracted by visual interpretation are shown in Fig. 5B, where the white part represents the building. Figures 5C and 5D show the extraction results using eCognition software and the extraction results of buildings using the proposed method. It can be seen from Fig. 5C that the extraction results of eCognition software misextract the spectral features of the bare land on the upper left side of the image, the road on the right lower side, and the small square in the lawn between buildings and other areas similar to buildings as buildings, and there is a serious misextraction phenomenon. In addition, the eCognition software failed to extract two low-rise buildings in the upper right side and the central region of the image, and failed to accurately extract the green part of the roof color of the building in the upper side of the image, resulting in serious missing extraction. It can be seen from Fig. 5D that the method in this article completely extracts 20 multi-story buildings with large scale and three low-rise buildings with small scale in the image, and extracts one building with very small scale in the lower area of the image (the pixel number is less than 100). In addition, based on the adjacent criterion of buildings and shadows and the inclusion criterion of buildings and building areas proposed in this article, the method eliminates the pseudo-building objects with similar spectral features, such as high brightness bare land, roads, cement ground. The extraction results are better than those of eCognition software. There are still some shortcomings in the extraction results of the experimental image, such as the failure to extract the shadowed part of the low-rise building in the middle of the image. In addition, for part of the green roof color on the building on the upper side of the image, the description of the spectral features of the building can be corrected according to the prior knowledge of the image.

Figure 5 Experiment 3 comparison of building extraction results.

Experiment 4 is a part of WorldView-2 satellite remote sensing image near Wuhan Wuhan University, as shown in Fig. 6A. The study area is the area with thin cloud cover under the original image. To eliminate the adverse effects of thin clouds on ground object extraction and enhance the details of buildings and shadows in the image, the image is homogenized before building extraction. The reference buildings manually extracted by visual interpretation are shown in Fig. 6B, where the white part represents the building. Figures 6C and 6D show the extraction results using eCognition software and the building extraction results using the proposed method, respectively. It can be seen from the figure that the proposed method completely extracts 16 multi-story buildings with large image scale and two low-rise buildings with small regional scale on the upper right side of the image. At the same time, based on the adjacent criterion of buildings and shadows, the proposed method eliminates the vacant land with similar spectral features in the middle region of the image. Figure 6C shows that the extraction results of eCognition software mistakenly extract the empty land similar to the spectral features of buildings in the central region of the image as buildings. In addition, eCognition software failed to extract two small-scale low-rise buildings in the upper right side of the image, and there was a phenomenon of missing extraction. In addition, due to the complexity of building roof detail information in Experiment 4’s image, the eCognition software failed to accurately extract some small targets on the roof, resulting in incomplete extraction of building objects, some building objects appeared as holes, or even fractured. Through comprehensive analysis, the overall effect of the proposed method on the extraction results of the two groups of experimental images is better than that of the eCognition software.

Figure 6 Experiment 4 comparison of building extraction results.

Evaluating indicators

Classification indicators

TP represents the prediction result as a positive sample, which is also the actual positive sample. FP represents the prediction result as a positive sample, but the actual sample is negative. TN represents the prediction result as a negative sample, which is also the actual negative sample. FN represents the prediction result as a negative sample, but the actual sample is positive. P′, P, N′, N, Total are explained as follows:

(15) P′=TP+FP

(16) P=TP+FN

(17) N′=FN+TN

(18) N=FP+TN

(19) Total=TP+FP+FN+TN

Producer accuracy, user accuracy and overall accuracy

Producer’s accuracy (PA) represents the ratio of correctly classified pixels of a certain category in the ground truth. User’s accuracy (UA) represents the ratio of correctly classified pixels of a certain category to the total number of pixels classified as that category. Overall accuracy (OA) represents the ratio of correctly classified pixels to the total number of pixels in the image. According to the confusion matrix for binary classification, PA, UA, and OA of the construction area are defined as:

(20) PA=TPP

(21) UA=TPP′

(22) OA=TP+TNTotal

Quantity disagreement and allocation disagreement

Quantity disagreement (QD) refers to the difference in quantity between the extracted results and the actual situation. Allocation disagreement (AD) refers to the difference in distribution between the extracted results and the actual situation, which occurs when the extracted results do not completely match the actual situation. Quantity disagreement and allocation disagreement are manifestations of overall error in terms of quantity and distribution, respectively. The sum of quantity disagreement and allocation disagreement represents the overall error.

In particular, for binary classification, the QD and AD can be calculated by the following equation:

(23) QD=|P−P′|+|N−N′|2×Total

(24) AD=min[P−TPTotal,P′−TPTotal]+min[N−TNTotal,N′−TNTotal]

The overall accuracy (OA) has the following relationship with QD and AD:

(25) OA=1−(QD+AD)

Comparative analysis

Analysis for shadow extraction results

Table 1 shows the accuracy of the shadow extraction results of the three methods for the two groups of experimental data. It can be seen from the table that for the first group of experiments, the extraction results of the proposed method are better than those of the other two methods in all indicators. The UA and PA are higher than 97 %, and the OA reaches 98.63 %. Compared to the other two methods, the method proposed in the article has lower AD and QD, both below 1 %. For the second group of experiments, UA and PA of the extraction results of the proposed method are about 90 %, and OA reaches 94.33 %. PA is only slightly inferior to the shadow extraction method based on continuous threshold, and all other indicators are better than the comparison method. However, UA of the extraction results of the second experiment image based on the continuous threshold method is only 54.72 %. It can be seen from Fig. 4D that the method has a large area of false extraction of the forest area on the upper side of the image, and UA is too low to meet the practical application requirements. Our QD can be as low as 0.78 %, and our method error is even lower, resulting in better extraction performance. In summary, the object-oriented shadow extraction method based on object feature knowledge base proposed in this article can effectively extract the shadow area in high-resolution RSI, and the extraction result is better than that of the traditional pixel-based extraction method.

Table 1 Evaluation of shadow extraction accuracy.

		UA%	PA%	OA%	AD%	QD%	
Experiment 1	Specthem ratio	89.47	91.17	93.94	5.47	0.59	
Continuous threshold	91.73	85.33	93.07	4.77	2.16	
Proposed	98.51	97.05	98.63	0.91	0.46	
Experiment 2	Specthem ratio	69.19	81.57	83.91	10.83	5.26	
Continuous threshold	54.72	89.40	75.14	6.23	18.63	
Proposed	91.45	89.02	94.33	4.89	0.78	

Analysis for building extraction results

Table 2 shows the accuracy of building extraction results using the proposed method and eCognition software for two groups of experimental images. It can be seen from the table that UA and PA of the proposed method for the extraction results of the two groups of experimental images are higher than 95 %, OA is higher than 97 %, and QD is lower than 1 %. In the two groups of experiments, OA of the extraction results using eCognition software was lower than 91 %, in which UA and PA of Experiment 3 were lower than 90 %, and UA and PA of Experiment 4 were lower than 80 %. In addition, AD of eCognition software for the extraction results of the two groups of experimental images is about 10 %, while AD of the extraction results of the proposed method is about 2 %. For two groups of experimental images, the accuracy of the proposed method is better than that of eCognition software in five indicators.

Table 2 Evaluation of building extraction accuracy.

		UA%	PA%	OA%	AD%	QD%	
Experiment 3	eCognition	80.69	89.44	87.98	7.94	4.07	
Proposed	97.23	95.4	97.25	2.04	0.71	
Experiment 4	eCognition	78.62	79.7	90.67	9.02	0.31	
Proposed	95.27	95.86	98.02	1.84	0.14	

Based on the comprehensive analysis of the visual effect and accuracy of two sets of experiments, the proposed object-oriented high-resolution RSI building extraction method under visual attention mechanism can effectively extract the buildings in high-resolution RSI, and the extraction results are better than the traditional object-oriented extraction method.

Conclusions

In high-resolution RSI and very high-resolution RSI, buildings often have high brightness, strong edges, obvious textures, and are usually accompanied by building shadows. We makes full use of the spectral features, shape features, texture features, edge features, context and topological relations of buildings in high-resolution RSI. Based on the visual attention mechanism drawing on information theory and the object-oriented image analysis method, we proposes an object-oriented high-resolution RSI building extraction method under visual attention mechanism by combining bottom-up low-level feature extraction of RSI with top-down prior knowledge guidance.

Additional Information and Declarations

Competing Interests

Author Contributions

Data Availability

The authors declare that they have no competing interests.

Xiaole Shen conceived and designed the experiments, performed the computation work, prepared figures and/or tables, and approved the final draft.

Chen Yu performed the experiments, analyzed the data, performed the computation work, prepared figures and/or tables, authored or reviewed drafts of the article, and approved the final draft.

Lin Lin performed the experiments, analyzed the data, prepared figures and/or tables, authored or reviewed drafts of the article, and approved the final draft.

Jinzhou Cao conceived and designed the experiments, authored or reviewed drafts of the article, and approved the final draft.

The following information was supplied regarding data availability:

The data and code are available at GitHub and Zenodo:

- https://github.com/sxxl86/buildingextraction.

- sxxl. (2023). sxxl86/buildingextraction: main (main). Zenodo. https://doi.org/10.5281/zenodo.8115357.

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
