# Peer review of "Object-oriented building extraction based on visual attention mechanism"

_PeerJ Computer Science, doi:10.7717/peerj-cs.1566_

## Round 0.1 · original submission · Minor Revisions

In this study, the authors proposed an object-oriented high-resolution remote sensing image building extraction algorithm. A total of three comments were received. All reviewer comments should be carefully considered and addressed.
Although two referees recommended minor revision, I have some concerns. Please highlight your main innovation in the revision. Necessary research gaps should be added. The submission template and citation format should be followed. My decision is minor revision.

Reviewer 1 ·

Basic reporting

This article presents an object-oriented building extraction based on the visual attention mechanism. The experiments performed on three typical remote sensing images demonstrate the superiority of the proposed method for extracting buildings from remote sensing images. However, there are still some necessary improvements that need to be made in the re-submitted version. The authors are recommended to carefully improve the writing and optimize the layout of section content.

The comments are listed as follows:

First, Figure 1 and Figure 2 should be clear and have sufficient resolution.

Second, we recommend using abbreviations for certain expressions, such as "remote sensing images" which can be replaced with "RSI". Additionally, some sentences are too long to read and understand. To make the article more manageable, we recommend breaking up long sentences into smaller, more concise chunks. Meanwhile, the methods section is redundant, which also needs to be clarified.

Finally, some necessary descriptions of the formula variables are absent. To ensure that readers can follow the mathematical formulas presented in the article, it is important to provide clear and comprehensive explanations of the variables used.

Experimental design

As a binary segmentation task, in order to highlight the results obtained by different methods, please use different RGB colours to label TPs, FPs, and FNs in Fig 3-6.

Validity of the findings

no comment

Additional comments

no comments.

Cite this review as

Reviewer 2 ·

Basic reporting

This manuscript proposed a method to extract buildings based on visual attention mechanism. It first generates the primary features, then BSI was proposed to combine textural features. Then building objects were extracted using BSI and refined by morphological operation. The paper is well organized.
Remarks for the paper:
1. More details about the experimental results should be included in the abstract section.
2. Previous methods about building extraction need to be added in introduction.

Experimental design

3. The author might briefly explain TP, P, TN, etc., in equations (22)(23)(24).
4. The language of this paper needs to be polished.

Validity of the findings

no comment

Additional comments

no comment

Cite this review as

·

Basic reporting

no comment

Experimental design

no comment

Validity of the findings

no comment

Additional comments

The paper proposed an object-oriented building extraction method based on visual attention mechanism. The method combined bottom-up primary features with top-down empirical knowledge. The method also used shadow information and built-up areas information to improve the accuracy. Experiments show the effectiveness of the method. The paper is well-structured, but still has some writing and formatting problem.
Comments for the paper:
1.References from recent years should be listed.
2.Figure 1 and Figure 2 are unsharp, and the flows are not clear enough.
3.The results of shadow extraction experiments should be quantitatively analyzed in more detail.
4.Some grammar and expression problems should be noted. For example, “Paper Algorithm” should be “Proposed Algorithm”.
5.Initial capitalization problems in some Figures and Tables should be noted.

---

## Round 0.2 · accepted · Accept

The authors have done a good job in revising the manuscript. Technical details are provided and explained. I congratulate the authors for the effort put into this paper! The manuscript is significantly improved; therefore, I recommend accepting it in its current form!